# Network Learning in Quadratic Games from Fictitious Plays

## Abstract

We study the ability of an adversary learning the underlying interaction network from repeated fictitious plays in linear-quadratic games. The adversary may strategically perturb the decisions for a set of action-compromised players, and observe the sequential decisions from a set of action-leaked players. Then the question lies in whether such an adversary can fully re-construct, or effectively estimate the underlying interaction structure among the players. First of all, by drawing connections between this network learning problem in games and classical system identification theory, we establish a series of results characterizing the learnability of the interaction graph from the adversary's point of view. Next, in view of the inherent stability and sparsity constraints for the network interaction structure, we propose a stable and sparse system identification framework for learning the interaction graph from full player action observations. We also propose a stable and sparse subspace identification framework for learning the interaction graph from partially observed player actions. Finally, the effectiveness of the proposed learning frameworks is demonstrated in numerical examples.

## 1 Introduction

Game theory has been playing a fundamental role in understanding how competition and rationality arise among decision makers, termed *players* (Von Neumann & Morgenstern (2007); Bicchieri (2004)). Players place individual decisions, for which the *payoffs* characterize their gains. The payoff of any player not only depends on her own decision, but also the decisions of all other players, leading to intrinsic competitions. The *Nash equilibrium* defines decision profiles at which any player observing her payoff and other players decisions, understands that any unilateral change of the decision could only upset her own payoff. As a result, at Nash equilibriums all players tend to stay at their current decisions, i.e., rationality takes place. The power of Nash equilibriums in explaining behavioral decisions, and the possibility of designing payoffs in shaping player decisions, have enabled game theory to be applied in a variety of disciplines solving problems ranging from route planning in transportation (Bianco et al. (2016)) and channel allocations in wireless communications (Niyato & Hossain (2008)) to online E-commerce (Sen & King (2003)) and security of machine learning (Barreno et al. (2010)).

The crucial insight in game-theoretical decision mechanisms, lies in the fact that players act according to their own payoff functions and other players' decisions, holding no knowledge about other players' payoffs. Since the payoffs might encode players' preferences, economic status, and interpersonal relations, etc., it is of interest to investigate when and how payoffs of players can be learned from observations of player actions.

### 1.1 Linear Quadratic Games

A game can be associated with a graph, where the players are represented by nodes, and the interdependency among the players in payoffs defines links. In a network game with $n$ players and linear-quadratic payoffs, each node (player) $i \in V := \{1, 2, \dots, n\}$ selects her action $\mathbf{x}_i \in \mathbb{R}$ to

maximize payoff $J_i$ described by

$$J_i = \alpha_i \mathbf{x}_i - \frac{1}{2}\mathbf{x}_i^2 + \sum_{j=1}^{n} g_{ij}\mathbf{x}_i\mathbf{x}_j. \tag{1}$$

In (1), the first two terms characterize the benefit of player $i$ by setting her own action $\mathbf{x}_i$, where the parameter $\alpha_i > 0$ is called the marginal benefit, capturing the level of selfishness of player $i$. The last term of this payoff function reflects the peer effect suffered by player $i$ from the actions of other players: if $g_{ij} > 0$, players $i$ and $j$ are strategic complements (friends or acquaintances); if $g_{ij} < 0$, players $i$ and $j$ are strategic substitutes (opponent or adversary); if $g_{ij} = 0$, there is no influence on player $i$ from player $j$.

Let $\mathbf{G} \in \mathbb{R}^{n \times n}$ be the matrix formed by the $g_{ij}$, i.e., the $ij$-entry of $\mathbf{G}$ is $g_{ij}$. The interaction graph $G = (V, E)$ underlying the game (1) is then defined as the induced graph of $\mathbf{G}$, where a directed link $(j, i) \in E$ if and only if $g_{ij} \neq 0$.

### 1.2 FICTITIOUS PLAYS AND ACTION-COMPROMIZED PLAYERS

For players participating in a game, Nash equilibriums can not be known or played immediately when the game starts since the payoff functions are held in private. Instead, in real-world the behaviors of players are better described in a sequential decision process (Littman (1994)). Let time be indexed at $k = 0, 1, 2, \ldots, T$, and let player $i$ hold decision $\mathbf{x}_i(k)$ for time $k$. Then, it is reasonable to assume that any rational player at any given $t$ will decide her next action as the decision that maximizes her payoff given other players' current decisions. This is known as *fictitious plays* or *best responses* (Fudenberg et al. (1998)). As a result, in repeated plays for the linear-quadratic game, the dynamics of $\mathbf{x}_i(k)$ obey

$$\mathbf{x}_i(k+1) = \arg\max_{\mathbf{x}_i} J_i(\mathbf{x}_i, \mathbf{x}_{-i}(k)) = \alpha_i + \sum_{j=1}^{n} g_{ij}\mathbf{x}_j(k), \ \ i = 1, \ldots, n. \tag{BR}$$

In practice, players might be influenced or manipulated by an adversary in their actions. We term such players as *action-compromised* players, indexed in the set $M \subseteq V$. Then $V \setminus M$ contains *benign* players who just follows (BR). For $i \in M$, we model the influence of the adversary as a perturbation $\mathbf{u}_i(k)$ over the best response. The fictitious plays of the players in the presence of the action-compromised players become

$$\mathbf{x}_i(k+1) = \arg\max_{\mathbf{x}_i} J_i(\mathbf{x}_i, \mathbf{x}_{-i}(k)) + \mathbf{I}_{\mathrm{M}}(i)\mathbf{u}_i(k)$$
$$= \alpha_i + \sum_{j=1}^{n} g_{ij}\mathbf{x}_j(k) + \mathbf{I}_{\mathrm{M}}(i)\mathbf{u}_i(k), \ \ i = 1, \ldots, n. \tag{p-BR}$$

where $\mathbf{I}_{\mathrm{M}}(i)$ is the indicator function: $\mathbf{I}_{\mathrm{M}}(i) = 1$ if $i \in M$; $\mathbf{I}_{\mathrm{M}}(i) = 0$ otherwise.

### 1.3 PROBLEM DEFINITION

For the adversary, it is easier and affordable to observe actions only from a number of *action-leaked* players. We let $O \subseteq V$ denote the group of players having their actions being observed. In this paper, we are interested in whether and how the underlying graph G (or equivalently, its adjacency matrix $\mathbf{G}$) is exposed to risks of being fully identified/learned by the adversary. To be precise, the adversary holds information

$$\mathcal{I} : \big\{\mathbf{u}_i(k), i \in M, k = 0, \ldots, T\big\} \bigcup \big\{\mathbf{x}_j(k), j \in O, k = 0, \ldots, T\big\}$$

with the $\mathbf{u}_i(k)$ and $\mathbf{x}_i(k)$ being produced by (p-BR). The problems of interest from the perspective of the adversary lie in

- *Learnability*: Whether it is possible to uniquely determine $\mathbf{G}$ from $\mathcal{I}$, perhaps with the help of strategically designed $\mathbf{u}_i(k), i \in M, k = 0, \ldots, T$.
- *Learning*: How to build effective learning frameworks for estimating $\mathbf{G}$ from $\mathcal{I}$, perhaps with prior structural information such as stability and sparsity.

### 1.4 CONTRIBUTIONS AND RELATED WORK

**Contributions**. The connection between the network learning problem in quadratic games and classical system identification theory is uncovered. The key insight is the observation that the unknown marginal payoffs do not play any role in the dynamics driving the difference between two consecutive player actions. As a result, we manage to establish a series of results characterizing the learnability of the interaction graph by the adversary: with full player action observations, the learnability is shown to be equivalent to certain observability conditions; with partial action observations, the learnability is shown to be determined by the unique solvability of an equation for Markov parameters. Next, noting the inherent stability and sparsity properties for the network interaction structure, we propose a stable and sparse system identification framework for learning the interaction graph from full player action observations, and a stable and sparse subspace identification framework for learning the interaction graph from partially observed player actions. Numerical examples validate the effectiveness of the proposed learning frameworks.

**Related Work**. Our paper is closely related to the recent studies on learning network games, such as Honorio & Ortiz (2015); Ghoshal & Honorio (2017); Leng et al. (2020); Garg & Jaakkola (2016; 2017); Ghoshal & Honorio (2018), where the behavioral actions are collected to recover the game graph. Particularly, Leng et al. (2020) studies a learning problem of linear quadratic games on networks, over which a number of independent games are played with all Nash equilibrium actions observed for learning the graph. We note that the learning frameworks in these works focus on static games where observations for learning are steady or Nash equilibrium actions, which is different from our considered setting of dynamic games where transient actions are observed during the decision-making process.

The considered learning problem is also related to the network inference problem in the fields of machine learning and signal processing. To handle such problem, several approaches have been proposed, differing in the applied models such as probabilistic graphical models (Friedman et al. (2008)), physically-motivated models (Gomez-Rodriguez et al. (2011)) and signal processing models (Dong et al. (2019)).

Technically, this work is built upon the results of linear system identification (Van Overschee & De Moor (2012); Ljung (1998)), in which measured input/output data are explored to build mathematical models of linear (network) systems, and even identify the system matrices or network structure under some structure and/or excitation conditions e.g., Bazanella et al. (2019); Yu et al. (2019); Shen et al. (2017). Recent advances of system identification include its applications to machine learning (Chiuso & Pillonetto (2019)), reinforcement learning (Ross & Bagnell (2012)) and deep learning (De la Rosa & Yu (2016)). In view of this, this paper can be regarded as a generalization of system identification approaches to network learning of games.

## 2 NETWORK LEARNABILITY FROM FICTITIOUS PLAYS

In this section, we investigate the learnability condition under which the network structure $\mathbf{G}$ can be uniquely determined by the adversary. The adversary may have access to the system dynamics (p-BR) for a single trajectory where the network game is played only once, or for multiple trajectories where the network game is independently played for a number of times.

### 2.1 LEARNABILITY BY SINGLE TRAJECTORY

For notational simplicity, we denote the aggregated action profile of all players and the overall injected perturbations at time $k$ by $\mathbf{x}(k) := [\mathbf{x}_1(k), \ldots, \mathbf{x}_n(k)]^\top \in \mathbb{R}^n$ and $\mathbf{u}(k) := [\mathbf{u}_i(k), i \in \mathrm{M}]^\top \in \mathbb{R}^m$, respectively. We also denote the cardinalities of the sets M and O as $m$ and $l$, respectively. Within these sets, the indexes of *action-compromised* and *action-leaked* players are sorted in ascending order with $\mathrm{M} := \{p_1, \ldots, p_m\}$ and $\mathrm{O} := \{q_1, \ldots, q_l\}$.

Introduce two matrices $\mathbf{B} \in \mathbb{R}^{n \times m}$ and $\mathbf{C} \in \mathbb{R}^{l \times n}$ to represent the sets of M and O. Specifically, the $(i, j)$-th entry of matrix $\mathbf{B}$ satisfies that

$$\mathbf{B}_{i,j} = \begin{cases} 1, & i = p_j; \\ 0, & \text{otherwise.} \end{cases} \tag{2}$$

The $(j, i)$-th entry of matrix $\mathbf{C}$ satisfies that

$$\mathbf{C}_{j,i} = \begin{cases} 1, & i = q_j; \\ 0, & \text{otherwise.} \end{cases} \tag{3}$$

From classical control theory (Wonham (1985)), the pair $(\mathbf{G}, \mathbf{B})$ is said to be *controllable* if the $n \times nm$ matrix (where the subscript $n$ denotes the number of block columns)

$$\mathcal{C}_n := \begin{bmatrix} \mathbf{B} & \mathbf{GB} & \ldots & \mathbf{G}^{n-1}\mathbf{B} \end{bmatrix}$$

has rank $n$; the pair $(\mathbf{G}, \mathbf{C})$ is said to be observable if the $nl \times n$ matrix (where the subscript $n$ denotes the number of block rows)

$$\mathcal{O}_n := \begin{bmatrix} \mathbf{C} \\ \mathbf{CG} \\ \vdots \\ \mathbf{CG}^{n-1} \end{bmatrix}$$

has rank $n$.

### 2.1.1 FULL PLAYER ACTION OBSERVATIONS

We present the following result showing that if the adversary has access to all players' actions but has no ability to add action perturbations, namely $\mathrm{O} = \mathrm{V}$ and $\mathrm{M} = \emptyset$, the learnability of the network structure $\mathbf{G}$ can be precisely characterized by a special form of observability.

**Theorem 1** *Assume* $\mathrm{O} = \mathrm{V}$ *and* $\mathrm{M} = \emptyset$. *Let the adversary has access to one single trajectory of* $\mathcal{I}$ *from (p-BR). Then* $\mathbf{G}$ *can be uniquely determined by the adversary for sufficiently large* $T$ *if and only if* $(\mathbf{G}, \mathbf{x}(1) - \mathbf{x}(0))$ *is controllable.*

In fact, we can further prove that $(\mathbf{G}, \mathbf{x}(1) - \mathbf{x}(0))$ is observable if and only if the characteristic polynomial of $\mathbf{G}$ coincides with the minimal polynomial of $\mathbf{G}$, and $\mathbf{x}(1) - \mathbf{x}(0)$ has nontrivial projections on to all generalized eigenvector of $\mathbf{G}$ (see more details in Proposition 1 of Appendix B). The next result characterizes the ability to learn $\mathbf{G}$ by feedback perturbations, namely $\mathbf{u}(k) = \mathbf{Kx}(k)$ for $\mathbf{K} \in \mathbb{R}^{m \times n}$.

**Theorem 2** *Assume* $\mathrm{O} = \mathrm{V}$ *and* $\mathrm{M} \neq \emptyset$. *Let the adversary has access to one single trajectory of* $\mathcal{I}$ *from (p-BR). Suppose* $\mathbf{u}(k)$ *is generated by* $\mathbf{u}(k) = \mathbf{Kx}(k)$. *Then there exists* $\mathbf{K}$ *such that* $\mathbf{G}$ *can be uniquely determined if the following conditions hold:*

*(i)* $\mathbf{0} \neq \mathbf{x}(1) - \mathbf{x}(0) \in \mathrm{Im}(\mathbf{B})$;

*(ii)* $(\mathbf{G}, \mathbf{B})$ *is controllable.*

### 2.1.2 PARTIAL PLAYER ACTION OBSERVATIONS

When $\mathrm{O}$ is only a subset of $\mathrm{V}$, the learnability of $\mathbf{G}$ by the adversary becomes a much more challenging question.

*Block Hankel Matrices.* For a time series, the block Hankel matrices provide a way of representing the time evolution in compact forms, given by an operator $\mathcal{H}$ that maps the time series into block matrices Van Overschee & De Moor (2012) (see Appendix A). Let $\mathbf{w}(k)$ denote the collection of the observed actions $\{\mathbf{x}_i(k), i \in \mathbf{O}\}$ generated from (p-BR) at time $k$. The block Hankel matrices of the sequences $\mathbf{W}_{0|T}, \mathbf{U}_{0|T}$ can be defined, respectively, leading to $\mathbf{W}_p, \mathbf{W}_f$ from $\mathcal{H}(\mathbf{W}_{0|T-1})$, $\mathbf{W}_p^+, \mathbf{W}_f^+$ from $\mathcal{H}(\mathbf{W}_{1|T})$, $\mathbf{U}_p, \mathbf{U}_f$ from $\mathcal{H}(\mathbf{U}_{0|T-1})$, and $\mathbf{U}_p^+, \mathbf{U}_f^+$ from $\mathcal{H}(\mathbf{U}_{1|T})$.

*Convolution Matrix.* The lower-triangular convolution matrix for the triplet $(\mathbf{G}, \mathbf{B}, \mathbf{C})$ is defined as

$$\mathcal{T}_n := \begin{bmatrix} 0 & 0 & \cdots & 0 \\ \mathbf{CB} & 0 & \cdots & 0 \\ \vdots & \ddots & \ddots & \vdots \\ \mathbf{CG}^{n-2}\mathbf{B} & \cdots & \mathbf{CB} & 0 \end{bmatrix}.$$

The reversed extended controllability matrix for a matrix pair $(\mathbf{G}, \mathbf{B})$ is defined as

$$\widehat{\mathcal{C}}_n := \begin{bmatrix} \mathbf{G}^{n-1}\mathbf{B} & \dots & \mathbf{GB} & \mathbf{B} \end{bmatrix}.$$

*Markov Parameter Equations.* For any $k$, $\mathbf{x}(k)$ can be represented as a linear combination of the past observations $\{\mathbf{x}_i(s), s = 0, \dots, k-1, i \in \mathrm{O}\}$ and the past perturbations $\{\mathbf{u}_j(s), s = 0, \dots, k-1, j \in \mathrm{M}\}$ with $s > 0$. Consequently, the elements in the first column in matrix $\mathcal{T}_n$ (termed Markov parameters) must satisfy the following equation (see Appendix D):

$$\Delta\mathbf{W}_f = \mathcal{O}_n\mathbf{G}^n\mathcal{O}_n^\dagger\Delta\mathbf{W}_p + \mathcal{O}_n\left[\widehat{\mathcal{C}}_n - \mathbf{G}^n\mathcal{O}_n^\dagger\mathcal{T}_n\right]\Delta\mathbf{U}_p + \mathcal{T}_n\Delta\mathbf{U}_f, \tag{4}$$

where $\Delta\mathbf{W}_f := \mathbf{W}_f^+ - \mathbf{W}_f$, $\Delta\mathbf{W}_p := \mathbf{W}_p^+ - \mathbf{W}_p$ and $\Delta\mathbf{U}_f, \Delta\mathbf{U}_p$ are defined similarly.

We are now ready to present the following result.

**Theorem 3** *Let the adversary has access to one single trajectory of $\mathcal{I}$ from (p-BR). Suppose $(\mathbf{G}, \mathbf{C})$ is observable and $T \geq 2n$. Then $\mathbf{G}$ can be uniquely determined by the adversary if and only if the Markov parameter equation* (4) *admits a unique solution with respect to $\mathbf{G}$.*

## 2.2 Learnability by Infinite Trajectories

The transfer function of the perturbed best-response (p-BR) is defined as $F_\mathbf{G}(z) := \mathbf{C}(z\mathbf{I} - \mathbf{G})^{-1}\mathbf{B}$. Here $z \in \mathbb{C}$, and $F_\mathbf{G}(z)$ captures the relation between $\mathbf{u}$ and $y_{\mathbf{u},\mathbf{x}_0,\mathbf{G}}$ in the frequency ($z$) domain. Inspired by the recent breakthrough on network system identification (van Waarde et al. (2021)), we present the following result. It asserts that either full observation of players' actions or full injected perturbations is necessary for the learnability of $\mathbf{G}$ when the adversary has infinitely trajectories.

**Theorem 4** *Let the adversary have access to infinitely many trajectories of $\mathcal{I}$ from (p-BR) subject to different initial player actions and perturbation sequences. Then the network structure $\mathbf{G}$ is uniquely learnable by the adversary if and only if*

$$F_\mathbf{G}(z) = F_{\tilde{\mathbf{G}}}(z) \text{ for } \tilde{\mathbf{G}} \in \mathbb{R}^{n \times n} \implies \mathbf{G} = \tilde{\mathbf{G}};$$

*or equivalently, if and only if at least one of the following conditions holds:*

  *(i)* $\operatorname{rank}\mathbf{C} = n$ *and* $(\mathbf{G}, \mathbf{B})$ *is controllable;*

  *(ii)* $\operatorname{rank}\mathbf{B} = n$ *and* $(\mathbf{G}, \mathbf{C})$ *is observable.*

## 3 Network Learning with Full Action Observations

In this section, we develop a framework for learning the network structure $\mathbf{G}$ from the information $\mathcal{I}$ generated by (p-BR) with full action observations (i.e., $\mathrm{O} = \mathrm{V}$).

### 3.1 The Framework: Stable and Sparse System Identification

We define $\mathbf{y}_i(k) := \mathbf{x}_i(k+1) - \mathbf{x}_i(k)$ and $\mathbf{v}_i(k) := \mathbf{u}_i(k+1) - \mathbf{u}_i(k)$. Then (p-BR) can be rewritten as

$$\mathbf{y}(k+1) = \mathbf{G}\mathbf{y}(k) + \mathbf{B}\mathbf{v}(k).$$

In practice, it is expected that the best response dynamics (BR) should converge to a Nash equilibrium (Jackson (2010)). This is guaranteed by the condition (Ballester et al. (2006)) that $\rho(\mathbf{G}) < 1$, where $\rho(\mathbf{G})$ is the spectral radius. Moreover, in practice, $\mathbf{G}$ is typically a sparse matrix.

Let $\mathcal{S}$ be the set of stable matrices in $\mathbb{R}^{n \times n}$, i.e., $\mathcal{S} := \{A \in \mathbb{R}^{n \times n} : \rho(A) < 1\}$. When observations of the $\mathbf{x}(k)$ (and thus, $\mathbf{y}(k)$) are subject to the influence of noises,

$$\mathbf{y}^{\mathrm{m}}(k) = \mathbf{y}(k) + \mathbf{e}_k$$

will be the actually observed actions, where $\mathbf{e}_k \in \mathbb{R}^d, k = 0, \ldots, T$ are stationary random noises with zero mean and co-variance $\mathbf{S}_n$. So the adversary has access to

$$Z = [\mathbf{y}^{\mathrm{m}}(1) \quad \mathbf{y}^{\mathrm{m}}(2) \quad \cdots \quad \mathbf{y}^{\mathrm{m}}(T)]; \tag{5}$$

$$Y = [\mathbf{y}^{\mathrm{m}}(0) \quad \mathbf{y}^{\mathrm{m}}(1) \quad \cdots \quad \mathbf{y}^{\mathrm{m}}(T-1)]; \tag{6}$$

$$V = [\mathbf{v}(0) \quad \mathbf{v}(1) \quad \cdots \quad \mathbf{v}(T-1)]. \tag{7}$$

We propose the following Stable Sparse System Identification (SSSI) for learning the network structure $\mathbf{G}$ from full player action observations:

$$\text{SSSI}: \quad \mathbf{G}_{\text{SSSI}} = \arg\inf_{\mathbf{G} \in \mathcal{S}} \frac{1}{2}||Z - \mathbf{G}Y - \mathbf{B}V||_F^2 + \theta||\mathbf{G}||_1. \tag{8}$$

### 3.2 An Information-Theoretic Algorithm

As $\mathcal{S}$ is an open and nonconvex set, solving (8) is numerically challenging. We propose to adopt the recently developed algorithm for stable system identification in Jongeneel et al. (2021), where an information-theoretic projection onto $\mathcal{S}$ was used to soften the computational complexity by solving Riccati equations. This leads to the following algorithm.

---

**Algorithm 1:** Information-theoretic Projection SSSI Algorithm

---

**Input:** $Z, Y, V, \mathbf{B}$
**Output:** Network graph structure $\mathbf{G}$
1 Solve the regularized least square solution

$$\hat{\mathbf{G}} = \arg\min_{\mathbf{G}} \frac{1}{2}||Z - \mathbf{G}Y - \mathbf{B}V||_F^2 + \theta||\mathbf{G}||_1; \tag{9}$$

2 Random generate $\delta \geq 0$;
3 Solve the algebraic Riccati equation (DARE) with the unique solution $\mathbf{P}_\delta$:

$$\mathbf{P} = \mathbf{I} + \hat{\mathbf{G}}^\top \mathbf{P}\hat{\mathbf{G}} - \hat{\mathbf{G}}^\top \mathbf{P}(\mathbf{P} + (2\delta\mathbf{S}_n)^{-1})^{-1}\mathbf{P}\hat{\mathbf{G}}_T; \tag{10}$$

4 Compute $\mathbf{L}_\delta = -(\mathbf{P}_\delta + (2\delta\mathbf{S}_n)^{-1})^{-1}\mathbf{P}_\delta\hat{\mathbf{G}}$;
5 Return $\mathbf{G}_{\text{SSSI}} = \hat{\mathbf{G}} + \mathbf{L}_\delta$.

---

## 4 Network Learning with Partial Actions

In this section, we develop a framework for learning the network structure $\mathbf{G}$ from the information $\mathcal{I}$ generated by (p-BR) with partial action observations, i.e., the set of players that the adversary can observer actions O is only a subset of the overall player set V.

### 4.1 Refined Markov Parameter Equation

Recall that $\mathbf{w}(k)$ is the vector representing the collection of the observed actions $\{\mathbf{x}_i(k), i \in \mathbf{O}\}$ generated from (p-BR) at time $k$. Defining $\mathbf{z}(k) = \mathbf{w}(k+1) - \mathbf{w}(k)$, $\mathbf{Z}_f, \mathbf{Z}_p, \mathbf{V}_f, \mathbf{V}_p$ as the block Hankel matrices that can be constructed from the sequences $\{\mathbf{z}(k), k = 0, \ldots, T-1\}$ and $\{\mathbf{v}(k), k = 0, \ldots, T-1\}$, the Markov parameter equation (4) can be equivalently reformulated as follows:

$$\mathbf{Z}_f = \mathcal{O}_n \mathbf{G}^n \mathcal{O}_n^\dagger \mathbf{Z}_p + \mathcal{O}_n \left[ \hat{\mathcal{C}}_n - \mathbf{G}^n \mathcal{O}_n^\dagger \mathcal{T}_n \right] \mathbf{V}_p + \mathcal{T}_n \mathbf{V}_f \tag{11}$$

$$:= \Phi_n \mathbf{Z}_p + \Psi_n \mathbf{V}_p + \mathcal{T}_n \mathbf{V}_f.$$

This refined Markov parameter equation directly connects the network structure $\mathbf{G}$ with the data accessible by the adversary.

### 4.2 Network Learning by Stable and Sparse Subspace Identification

Again, when the observations of the players' actions $\mathbf{w}(k)$ (and thus, $\mathbf{z}(k)$) are masked by noises in practice, $\mathbf{z}^{\mathrm{m}}(k) = \mathbf{z}(k) + \mathbf{e}_k$ will be the actually observed action differences. The adversary has

access to

$$\begin{bmatrix} \mathbf{Z}_p^{\mathrm{m}} \\ \mathbf{Z}_f^{\mathrm{m}} \end{bmatrix} = \mathcal{H}\big(\mathbf{Z}_{0|T-1}^{m}\big), \quad \begin{bmatrix} \mathbf{V}_p \\ \mathbf{V}_f \end{bmatrix} = \mathcal{H}\big(\mathbf{V}_{0|T-1}\big).$$

We are now ready to propose the following Stable and Sparse Subspace Identification (SSSubI) learning framework.

---

**Algorithm 2:** Network Learning by Stable and Sparse Subspace Identification

---

**Input:** $\mathbf{Z}_p^{\mathrm{m}}, \mathbf{Z}_f^{\mathrm{m}}, \mathbf{V}_p, \mathbf{V}_f, \mathbf{C}, \mathbf{B}$
**Output:** Network graph structure $\mathbf{G}$

1 From (11) carry out classical subspace identification [Ref], and produce $\mathcal{T}_n^\star$ as an optimal estimate for $\mathcal{T}_n$;
2 Take $\mathbf{M}_l^\star$ as the $(l+2, 1)$-th block entries for $\mathcal{T}_n^\star$ for $l = 0, \ldots, n-2$, and then solve

$$\mathbf{G}_{\mathrm{SSSubI}} = \arg\min_{\mathbf{G} \in \mathcal{S}} \sum_{l=1}^{n-2} ||\mathbf{C}\mathbf{G}^l\mathbf{B} - \mathbf{M}_l^\star||_F^2 + \gamma ||\mathbf{G}||_1;$$

3 Return $\mathbf{G}_{\mathrm{SSSubI}}$.

---

When solving $\mathbf{G}_{\mathrm{SSSubI}}$, again we can first solve

$$\bar{\mathbf{G}} = \arg\min_{\mathbf{G} \in \mathbb{R}^{n \times n}} \sum_{l=1}^{n-2} ||\mathbf{C}\mathbf{G}^l\mathbf{B} - \mathbf{M}_l^\star||_F^2 + \gamma ||\mathbf{G}||_1, \tag{12}$$

and then use the information-theoretic projection in Jongeneel et al. (2021) to project $\bar{\mathbf{G}}$ onto $\mathcal{S}$.

## 5 Numerical Validations

In this section, we provide numerical examples to validate the two proposed learning frameworks. For reproduction of the reported results, all source codes have been provided in the **supplementary material**.

### 5.1 Full Action Observations: Stable and Sparse System Identification

In this section, we examine the performance of Algorithm 1 on three types of synthetic networks generated using the Erdős–Rényi (ER), the Watts–Strogatz (WS) and the Barabási–Albert (BA) models. The networks under evaluation have $n = 100$ nodes.

**Network setup**. In the ER graph, each link takes place with probability $p_{er} = 0.1$ independently with all the other links; in the WS graph, each node's average degree and the random rewiring process is set to be is $k_{ws} = 5$ and $p_{ws} = 0.2$, respectively; in the BA graph, a new node at each time step is created to connect to $m_{ba} = 2$ existing nodes via preferential attachment. After the realization of each graph, a nonzero random number is selected between $-5$ and $5$ as the weight for each link. Each entry of the adjacency matrix $\mathbf{G}$ is then divided by its largest absolute value of its eigenvalues to ensure the stability $\mathbf{G}$.

**Data synthesis**. A set M representing action-compromised players is selected and a matrix $\mathbf{B}$ is created according to (2). We then generate $\mathbf{x}(0)$, $\mathbf{u}(k), k = 0, \cdots, T$, and $\boldsymbol{\alpha}$ by considering $\mathbf{x}(0), \mathbf{u}(k), \boldsymbol{\alpha} \sim \mathcal{N}(0, \mathbf{I})$, and simulate the dynamics (p-BR) to obtain $\mathbf{x}(k), k = 1, \ldots, T$. The observation noises follow a normal distribution $\mathbf{e}_k \sim \xi * \mathcal{N}(0, \mathbf{I})$ where $\xi$ represents the noise intensity level. Upon Eq. (5)-(7), we finally obtain three matrices $Z, Y, V$.

**Experiment**. We apply Algorithm 1 to the respective settings and compare the outcomes against the ground truth by relative error

$$\mathrm{Err} := \frac{||\mathbf{G}_{\mathrm{SSSI}} - \mathbf{G}_{\mathrm{Truth}}||_F}{||\mathbf{G}_{\mathrm{Truth}}||_F}. \tag{13}$$

We also implement two other baseline approaches: the stable least square (SLS) and stable $L_2$-regularized least square (SL2LS), which are described in Appendix F.1. For SSSI and SL2LS, we

provide the results using the parameters $\theta$ and $\beta$ that yield the greatest average performance across 50 randomly generated graph instances.

**Results.** First and foremost, we are to discover how the the trajectory length affects the learning performance. We fix noise level to be $\xi = 0.1$, use 6 different values of $T = \{15, 20, 25, 30, 35, 40\}$ and follow the aforementioned data generation process. The learning performance of the three methods versus trajectory length on the ER, WS, BA networks is illustrated in Fig. 1. We next examine the robustness of the three methods in the face of various levels of noise during the observations. Let the trajectory length be $T = 40$ and the noise intensity level take values in $\xi = \{0.05, 0.1, 0.15, 0.2, 0.25, 0.3\}$. The learning performance of the three methods versus noise intensities on the ER, WS, BA networks is shown in Fig. 2. Clearly in terms of reconstructing the weights for the groundtruth links, the SSSI outperforms the other two baseline methods significantly. Moreover, the BA networks appear to be more difficult to learn under SSSI in all cases, compared to the ER graphs and WS graphs.

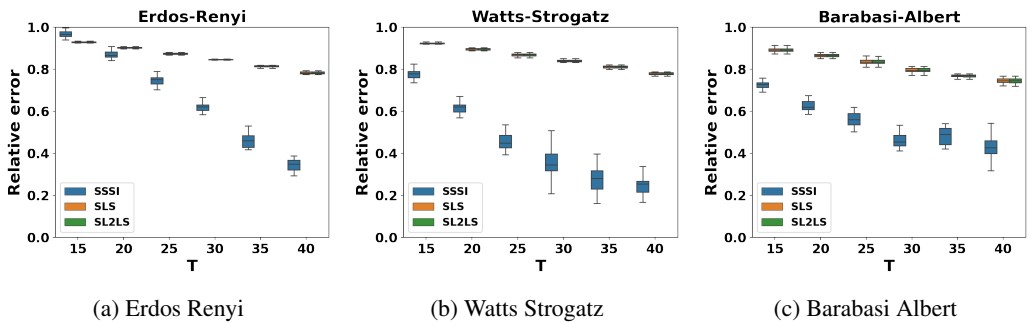

(a) Erdos Renyi   (b) Watts Strogatz   (c) Barabasi Albert

Figure 1: The relative errors from SSSI, SLS, SL2LS versus trajectory length over the ER, WS and BA networks.

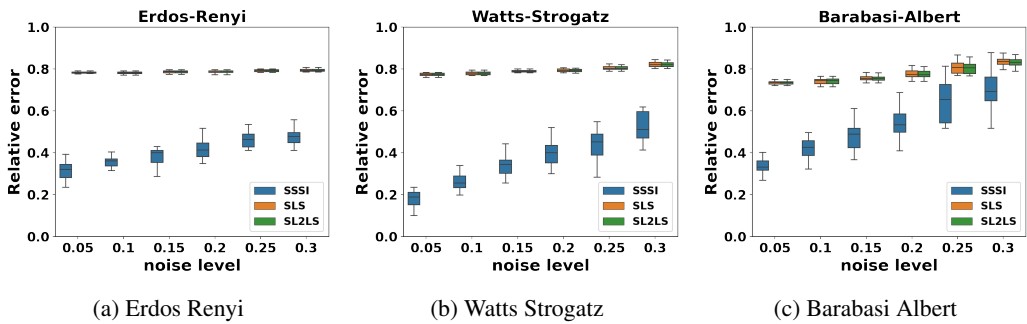

(a) Erdos Renyi   (b) Watts Strogatz   (c) Barabasi Albert

Figure 2: The relative errors from SSSI, SLS, SL2LS versus noise intensity level over the ER, WS and BA networks.

## 5.2 Network Learning with Partial Actions: Subspace Identification

In this section, we evaluate the performance of Algorithm 2 on the ER, WS and BA networks. In particular, since the Step 1 subspace identification has been an established approach (Van Overschee & De Moor (2012)), we focus our attention on the effectiveness of Step 2. The networks used for experiments are of the size of 10 nodes.

**Network setup.** We follow the network setup process in Section 5.1 to obtain the adjacency matrix **G**, where $p_{er} = 0.2, k_{ws} = 2, p_{ws} = 0.2$ and $m_{ba} = 1$.

**Data synthesis.** Two sets $O, M$ representing the action-leaked and action-compromised players are selected, from which $\mathbf{C}, \mathbf{B}$ are obtained based on (3) and (2). We compute $\mathbf{M}_l^\star, l = 0, \dots, n - 2$, directly from the ground truth of $\mathbf{G}, \mathbf{C}, \mathbf{B}$.

**Experiments.** The metric for learning performance evaluation continues to be the relative error described in (13). The benchmark is set to be stable subspace identification (SSubl) described in Appendix F.2, which is obtained by removing the the $L_1$ regularization from (SSSubl). We report the results using $\gamma$ that leads to the best average performance across 20 randomly created graph instances.

**Results.** The relative errors from SSSubl and SSubl over the ER, WS and BA networks are depicted in Fig. 3. We see that SSSubl indeed provides better accuracy than SSubl in inferring the network structure in all network types. Again, the BA networks are apparently more difficult to learn compared to ER and WS networks.

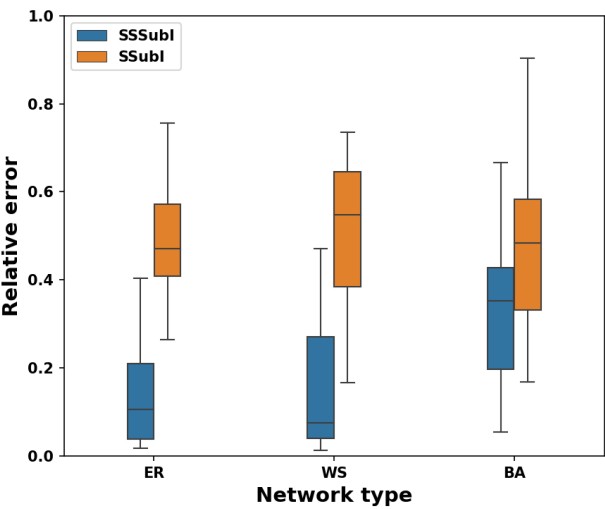

Figure 3: The relative errors from SSSubl and SSubl over the ER, WS and BA networks.

## 6 CONCLUSIONS

We have studied the problem of learning the interaction network structure from dynamic played actions of the players from the perspective of an adversary, who may strategically perturb the decisions for a set of action-compromised players, and observe the sequential decisions from a set of action-leaked players. Results characterizing the learnability of the interaction graph by the adversary were established, where key insights came from classical system identification theories. Two learning frameworks were proposed for fully and partially player actions, respectively, for which numerical examples validate their usefulness. This work opened the doors for learning the underlying interaction graph from fictitious plays in network games. Future work may include extensions of this line of research to network games with general payoff functions, and applications of the work to the real world.

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

## A BLOCK HANKEL MATRICES

For a time sequence $\mathbf{s}(k) \in \mathbb{R}^n, k = 0, \ldots, T$, a collection of samples from time $k$ to time $k + l$ is denoted as $\mathbf{S}_{k|k+l} := [\mathbf{s}(k) \quad \mathbf{s}(k+1) \quad \cdots \quad \mathbf{s}(k+l)]$. Take $h = T + 1 - 2n$. A block Hankel matrix of the sequence $\mathbf{S}_{0|T-1}$ is defined as

$$
\mathcal{H}(\mathbf{S}_{0|T-1}) := \left[ \begin{array}{ccccc}
\mathbf{s}(0) & \mathbf{s}(1) & \mathbf{s}(2) & \cdots & \mathbf{s}(h-1) \\
\mathbf{s}(1) & \mathbf{s}(2) & \mathbf{s}(3) & \cdots & \mathbf{s}(h) \\
\vdots & \vdots & \vdots & \ddots & \vdots \\
\mathbf{s}(n-1) & \mathbf{s}(n) & \mathbf{s}(n+1) & \cdots & \mathbf{s}(n+h-2) \\
\hline
\mathbf{s}(n) & \mathbf{s}(n+1) & \mathbf{s}(n+2) & \cdots & \mathbf{s}(n+h-1) \\
\mathbf{s}(n+1) & \mathbf{s}(n+2) & \mathbf{s}(n+3) & \cdots & \mathbf{s}(n+h) \\
\vdots & \vdots & \vdots & \ddots & \vdots \\
\mathbf{s}(2n-1) & \mathbf{s}(2n) & \mathbf{s}(2n+1) & \cdots & \mathbf{s}(T-1)
\end{array} \right]
$$
$$
:= \left[ \begin{array}{c} \mathbf{S}_p \\ \hline \mathbf{S}_f \end{array} \right]
$$

The block Hankel matrix of $\mathbf{S}_{1|T}$ is defined as

$$
\mathcal{H}(\mathbf{S}_{1|T}) := \left[ \begin{array}{c} \mathbf{S}_p^+ \\ \hline \mathbf{S}_f^+ \end{array} \right].
$$

## B PROOF OF THEOREM 1

Regarding the action dynamics described in (p-BR), we define the action difference and the perturbation difference as $\mathbf{y}_i(k) := \mathbf{x}_i(k+1) - \mathbf{x}_i(k)$ and $\mathbf{v}_i(k) := \mathbf{u}_i(k+1) - \mathbf{u}_i(k)$ respectively, in order to eliminate the effect of marginal benefit $\alpha_i$ in learning graph structure $\mathbf{G}$.

Adding the observation ports to the action dynamics (p-BR), we arrive at the compact form of an input-output linear invariant system:

$$
\mathbf{y}(k+1) = \mathbf{G}\mathbf{y}(k) + \mathbf{B}\mathbf{v}(k), \tag{14}
$$
$$
\mathbf{z}(k) = \mathbf{C}\mathbf{y}(k),
$$

in which the state $\mathbf{y}(k) := [\mathbf{y}_1(k), \ldots, \mathbf{y}_n(k)]^\top$, the input signal $\mathbf{v}(k) \in \mathbb{R}^m$, $\mathbf{B} \in \mathbb{R}^{n \times m}$ and $\mathbf{C} \in \mathbb{R}^{l \times n}$. The nonzero entries of matrices $\mathbf{B}$ and $\mathbf{C}$ correspond to the set of *action-compromised* and *action-leaked* players, respectively. Hence, the adversary has a trajectory of the input/output signal of this system (14): $\mathcal{I} := \{\mathbf{v}(k), \mathbf{z}(k), 0 \le k \le T\}$.

When $O = V$ and $M = \emptyset$, (14) is degenerated as

$$
\mathbf{z}(k+1) = \mathbf{y}(k+1) = \mathbf{G}\mathbf{y}(k).
$$

Then, the single trajectory satisfies

$$
Z = \mathbf{G}Y,
$$

where $Z = [\mathbf{y}(1), \mathbf{y}(2), \ldots, \mathbf{y}(T)] \in \mathbb{R}^{n \times T}$ and $Y = [\mathbf{y}(0), \mathbf{x}(2), \ldots, \mathbf{y}(T-1)] \in \mathbb{R}^{n \times T}$.

This equation can be decoupled as several linear equations: $Y^\top [\mathbf{G}^\top]_i = [Z^\top]_i, \forall i$ with $[\cdot]_i$ representing the $i$-th column. Then, $\mathbf{G}$ is uniquely constructable from the trajectory $\mathcal{T}$ if and only if each of these linear equations has a unique solution, i.e., $\text{rank}(Y) = \text{rank}([Y^\top, [Z^\top]_i]) = n, \forall i$.

Note that $Y = [\mathbf{y}(0), \mathbf{y}(1), \ldots, \mathbf{y}(T-1)] = [\mathbf{y}(0), \mathbf{G}\mathbf{y}(0), \ldots, \mathbf{G}^{T-1}\mathbf{y}(0)]$, of which the vectors span a $\mathbf{G}$-cyclic subspace of $\mathbb{R}^n$, denoted by $H(\mathbf{y}(0); \mathbf{G})$. This subspace is an invariant subspace for $\mathbf{G}$, in the sense that $\mathbf{G}H(\mathbf{y}(0); \mathbf{G}) \subseteq H(\mathbf{y}(0); \mathbf{G})$, which validates $\text{rank}(Y) = \text{rank}([Y^\top, [Z^\top]_i])$. Moreover, the condition $\text{rank}(Y) = n$ holds if and only if $(\mathbf{G}, \mathbf{y}(0))$ is controllable.

**Proposition 1** *The following statements are equivalent:*

  *1. $(\mathbf{G}, \mathbf{y}(0))$ is controllable;*

2. *The characteristic polynomial of* $\mathbf{G}$ *coincides with the minimal polynomial of* $\mathbf{G}$*, and* $\mathbf{y}(0)$ *has nontrivial projections on to all generalized eigenvector of* $\mathbf{G}$*.*

**Proof:** We first prove "1 $\Longrightarrow$ 2". Suppose that (i). the minimal polynomial of $\mathbf{G}$, termed $m(t)$, is not equal to its characteristic polynomial, termed $\kappa(t)$; or (ii). $\mathbf{y}(0)$ has zero projections on to one generalized eigenvector of $\mathbf{G}$.

For the pair $(\mathbf{G}, \mathbf{y}(0))$, we form the sequence of vectors

$$\mathbf{y}(0), \mathbf{G}\mathbf{y}(0), \mathbf{G}^2\mathbf{y}(0), \dots$$

and they reside in a $\mathbf{G}$-invariant subspace $\mathbb{P}$ with dimension $p$, where $p$ is the degree of the *minimal polynomial for vector* $\mathbf{y}(0)$ with the linear operator $\mathbf{G}$, termed $\zeta_{\mathbf{y}(0)}(t)$. Moreover, $\zeta_{\mathbf{y}(0)}(t)$ divides $m(t)$. Interested readers are suggested to (MacDuffee, 2012, Section VII-4) for the definition of *annihilating polynomial* and its property.

If (i) holds, i.e., $m(t) \neq \kappa(t)$, then the degree of $m(t)$ is less than $n$ as $m(t)$ divides $\kappa(t)$, and further the degree of $\zeta(t)$ satisfies $p < n$. Then $\mathrm{rank}([\mathbf{y}(0), \mathbf{G}\mathbf{y}(0), \dots]) = p < n$ and $(\mathbf{G}, \mathbf{y}(0))$ is uncontrollable. Obviously, if (ii) holds, the dimension of the $\mathbf{G}$-invariant subspace $\mathbb{P}$ satisfies $p < n$ and further $(\mathbf{G}, \mathbf{y}(0))$ is uncontrollable.

We next prove "2 $\Longrightarrow$ 1". Denote the set of distinct eigenvalues of $\mathbf{G}$ as $\sigma(\mathbf{G})$ with cardinality $g$. Given an eigenvalue $\lambda_i$, its geometric multiplicity, termed $\mu_i$, is equal to 1 since $m(t) = \kappa(t)$, and its algebraic multiplicity is equal to its multiplicity in the minimal polynomial $m(t)$ is denoted by $r_i$ with $\sum_{i=1}^g r_i = n$.

Let $e_{\lambda_i, j}$ denote the generalized eigenvectors of rank $j$ ($1 \le j \le r_i$) corresponding to $\mathbf{G}$ and its eigenvalue $\lambda_i \in \sigma(\mathbf{G})$. From statement 2, we have that

$$\mathbf{y}(0) = \sum_{i=1}^g E_i := \sum_{i=1}^g \alpha_{\lambda_i, r_i} e_{\lambda_i, r_i} + \alpha_{\lambda_i, r_i - 1} e_{\lambda_i, r_i - 1} + \cdots + \alpha_{\lambda_i, 1} e_{\lambda_i, 1}$$

with nonzero $\alpha_{\lambda_i, j}, i = 1, \dots, g, j = 1, \dots, r_i$. According to the definition of generalized eigenvector, we have

$$E_i := \sum_{j=1}^{r_i} \alpha_{\lambda_i, j} e_{\lambda_i, j} = [\sum_{j=1}^{r_i} \alpha_{\lambda_i, j} (\mathbf{G} - \lambda_i I)^{r_i - j}] e_{\lambda_i, r_i} := \mathbf{D} e_{\lambda_i, r_i}$$

with nonzero matrix $\mathbf{D}$. Note that $\mathbf{D}$ and $\mathbf{G}$ commute, and hence the *minimal polynomial of vector* $E_i$, termed $\zeta_{E_i}(t)$, is equal to the *minimal polynomial of vector* $e_{\lambda_i, r_i}$, termed $\zeta_{e_{\lambda_i, r_i}}(t) = (t - \lambda_i)^{r_i}$.

Since the *minimal polynomials for vectors* $E_i$ and $E_j$ with $j \neq i$ coprime, according to (MacDuffee, 2012, Section VII-4, Lemma in Page 181), the *minimal polynomials for vectors* $\mathbf{y}(0)$ is equal to the product of $\zeta_{E_i}(t), i = 1, \dots, g$. Hence, the degree of $\zeta_{\mathbf{y}(0)}$ is $p = \sum_{i=1}^g r_i = n$, which implies the controllability of the pair $(\mathbf{G}, \mathbf{y}(0))$. ∎

## C PROOF OF THEOREM 2

When $\mathrm{O} = \mathrm{V}$ and $\mathrm{M} \neq \emptyset$, (14) is degenerated as

$$\mathbf{z}(k+1) = \mathbf{y}(k+1) = \mathbf{G}\mathbf{y}(k) + \mathbf{B}\mathbf{v}(k).$$

Setting $\mathbf{v}(k) = \mathbf{K}\mathbf{y}(k)$, there holds

$$\mathbf{y}(k+1) = (\mathbf{G} + \mathbf{B}\mathbf{K})\mathbf{y}(k).$$

According to Theorem 1, $\mathbf{G}$ can be uniquely determined by the adversary from one single trajectory $\mathcal{I}$ if and only if $(\mathbf{G} + \mathbf{B}\mathbf{K}, \mathbf{y}(0))$ is controllable.

Then, the proof of Theorem 2 can be complemented by the following lemma (Wonham, 1985, Lemma 2.2), and the proof is omitted here.

**Lemma 1** *Let* $\mathbf{0} \neq \mathbf{y}(0) \in Im(\mathbf{B})$*. If* $(\mathbf{G}, \mathbf{B})$ *is controllable, there exists* $\mathbf{K}$ *such that* $(\mathbf{G} + \mathbf{B}\mathbf{K}, \mathbf{y}(0))$ *is controllable.*

## D    PROOF OF THEOREM 3

Consider the input sequence $\mathbf{V}_{0|T-1}$ and output one $\mathbf{Z}_{0|T-1}$ of (14), and construct the corresponding block Hankel matrices, as introduced in Section 2.1.2, termed $\mathbf{V}_p, \mathbf{V}_s, \mathbf{Z}_p, \mathbf{Z}_f$. Then, we can reformulate the state-space model in (14) as follows:

$$\mathbf{Z}_f = \mathcal{O}_n \mathbf{Y}_{n|T-n} + \mathcal{T}_n \mathbf{V}_f, \tag{15}$$

where the observability matrix $\mathcal{O}_n$ and the convolution matrix $\mathcal{T}_n$ are defined in Section 2.1.2, respectively. Throughout the article, it is stipulated that the dimension parameter $T+1 \gg 3n$ such that the Hankel matrices $\mathbf{Z}_f$ and $\mathbf{V}_f$ have more columns than rows. Note that in (15) the term $\mathbf{Y}_{n|T-n}$ is unknown to the adversary. Hence, we hope to construct $\mathbf{Y}_{n|T-n}$ from the past input-output data streams, and utilize it to establish an equation of $\mathbf{G}$ totally based on the input-output data streams.

First, consider the past output sequences constructed as follows

$$\mathbf{Z}_p = \mathcal{O}_n \mathbf{Y}_{0|T-2n} + \mathcal{T}_n \mathbf{V}_p,$$

similar to (15).

When the observability matrix $\mathcal{O}_n$ has full column rank, the state sequence $\mathbf{Y}_{n|T-n}$ can be represented as

$$\mathbf{Y}_{n|T-n} = \mathbf{G}^n \mathbf{Y}_{0|T-2n} + \hat{\mathcal{C}}_n \mathbf{V}_p$$
$$= \mathbf{G}^n \mathcal{O}_n^\dagger \mathbf{Z}_p + \left[ \hat{\mathcal{C}}_n - \mathbf{G}^n \mathcal{O}_n^\dagger \mathcal{T}_n \right] \mathbf{V}_p,$$

where $\hat{\mathcal{C}}_n$ is the reversed controllability matrix.

Replacing $\mathbf{Y}_{n|T-n}$ in (15) by the above representation, we obtain that

$$\mathbf{Z}_f = \mathcal{O}_n \mathbf{G}^n \mathcal{O}_n^\dagger \mathbf{Z}_p + \mathcal{O}_n \left[ \hat{\mathcal{C}}_n - \mathbf{G}^n \mathcal{O}_n^\dagger \mathcal{T}_n \right] \mathbf{V}_p + \mathcal{T}_n \mathbf{V}_f \tag{16}$$

By now, the adversary can construct (16) from the single trajectory $\mathcal{I}$. Then, the sufficiency and necessity of Theorem 3 can be easily derived from (16).

## E    DEFINITION OF INDISTINGUISHABLITY AND PROOF OF THEOREM 4

By introducing the action and perturbation difference, we obtain the state-space form (14) of the perturbed game play (p-BR). Let $\mathcal{Z}_{\mathbf{v},\mathbf{y}_0,\mathbf{G}}(\cdot)$ and $\mathcal{Z}_{\mathbf{v},\tilde{\mathbf{y}}_0,\tilde{\mathbf{G}}}(\cdot)$ denote the trajectories of $\mathbf{z}(k), k = 0, 1, \ldots$ of two different systems of the form (14), where the subscripts emphasize the dependence on the perturbation difference $\mathbf{v}(\cdot)$, the initial states $\mathbf{y}_0, \tilde{\mathbf{y}}_0$, and the topology matrix $\mathbf{G}, \tilde{\mathbf{G}}$, respectively. We introduce the following definition.

**Definition 1** *The topology matrices $\mathbf{G}$ and $\tilde{\mathbf{G}}$ are indistinguishable if there exist initial conditions $\mathbf{y}_0, \tilde{\mathbf{y}}_0 \in \mathbb{R}^n$ such that*

$$\mathcal{Z}_{\mathbf{v},\mathbf{y}_0,\mathbf{G}}(\cdot) = \mathcal{Z}_{\mathbf{v},\tilde{\mathbf{y}}_0,\tilde{\mathbf{G}}}(\cdot)$$

*for all injected perturbation difference $\mathbf{v}$.*

Clearly, the indistinguishability defined above points to infinite trajectories of $\mathcal{I}$ as it is concerned with *all possible* initial player actions and perturbation sequences. Obviously, the topology of the linear-quadratic network game is said to be *learnable* if $\mathbf{G}$ and $\tilde{\mathbf{G}}$ are distinguishable for all real $\mathbf{G} \neq \tilde{\mathbf{G}}$.

To prove Theorem 4, we give the following two statements[1]:

**Lemma 2** *A pair of parameter values $(\mathbf{G}, \tilde{\mathbf{G}})$ is indistinguishable if and only if the Markov parameters satisfying:*

$$\mathbf{C}\mathbf{G}^l\mathbf{B} = \mathbf{C}\tilde{\mathbf{G}}^l\mathbf{B}, l = 0, 1, 2, \cdots. \tag{17}$$

---

[1]The proofs of Lemma 2 and Lemmas 3-4 are inspired by and adopted from the proofs for system identifiability in Grewal & Glover (1976) and van Waarde et al. (2021), respectively.

**Proof:** We first prove the sufficiency. If (17) holds, then the output trajectories of two systems (14) associated with $(\mathbf{G}, \tilde{\mathbf{G}})$ satisfy that when $\mathbf{y}_0 = \tilde{\mathbf{y}}_0$,

$$\mathcal{Z}_{\mathbf{v}, \mathbf{y}_0, \mathbf{G}}(k) := \mathbf{z}(k) = \mathbf{C}\mathbf{G}^k \mathbf{y}_0 + \sum_{t=0}^{k-1} \mathbf{C}\mathbf{G}^t \mathbf{B} \mathbf{v}(k-1-t) = \mathcal{Z}_{\mathbf{v}, \tilde{\mathbf{y}}_0, \tilde{\mathbf{G}}}(k),$$

for all $\mathbf{v}(k) \in \mathbb{R}^m$ and for $k \geq 0$. According to Definition 1, the topology matrices $\mathbf{G}$ and $\tilde{\mathbf{G}}$ are indistinguishable.

The necessity is proved as follows. Assume $(\mathbf{G}, \tilde{\mathbf{G}})$ are indistinguishable, then there exist initial conditions $\mathbf{y}_0, \tilde{\mathbf{y}}_0$ s.t.

$$\mathbf{C}\mathbf{G}^k \mathbf{y}_0 + \sum_{t=0}^{k-1} \mathbf{C}\mathbf{G}^t \mathbf{B} \mathbf{v}(k-1-t) = \mathbf{C}\tilde{\mathbf{G}}^k \tilde{\mathbf{y}}_0 + \sum_{t=0}^{k-1} \mathbf{C}\tilde{\mathbf{G}}^t \mathbf{B} \mathbf{v}(k-1-t)$$

holds for all input functions $\mathbf{v}$ and for all $k \geq 0$. It implies that

$$\mathbf{C}\mathbf{G}^k \mathbf{y}_0 = \mathbf{C}\tilde{\mathbf{G}}^k \tilde{\mathbf{y}}_0, \forall k \geq 0, \tag{18}$$

and

$$\sum_{t=0}^{k-1} [\mathbf{C}\mathbf{G}^t \mathbf{B} - \mathbf{C}\tilde{\mathbf{G}}^t \mathbf{B}] \mathbf{v}(k-1-t) \equiv 0. \tag{19}$$

Hence, if $\mathbf{y}_0$ and $\tilde{\mathbf{y}}_0$ are set as zero vectors, then (18) holds for all $k$, which implies the existence of $\mathbf{y}_0$ and $\tilde{\mathbf{y}}_0$. Since $\mathbf{v}(\cdot) \in \mathbb{R}^m$ has a nonempty interior, (19) means that

$$\mathbf{C}\mathbf{G}^l \mathbf{B} = \mathbf{C}\tilde{\mathbf{G}}^l \mathbf{B}, \ \forall l = 0, 1, \dots.$$

Hence, the indistinguishability implies that at different values of the parameter $\mathbf{G}$, the Markov parameters of the system $\mathbf{C}\mathbf{G}^k \mathbf{B}$ are identical. ∎

**Lemma 3** *If the topology* $\mathbf{G}$ *is learnable, then* $(\mathbf{G}, \mathbf{B})$ *is controllable and* $(\mathbf{G}, \mathbf{C})$ *is observable.*

**Proof:** Suppose $(\mathbf{G}, \mathbf{B})$ is uncontrollable. Let $v \in \mathbb{R}^n$ be a nonzero vector in the uncontrollable subspace of $(\mathbf{G}, \mathbf{B})$, i.e.,

$$v^\top \mathbf{G}^l \mathbf{B} = 0, \ \forall k = 0, 1, \dots$$

Given a topology matrix $\mathbf{G}$, we can construct $\tilde{\mathbf{G}} = \mathbf{G} + vv^\top$, satisfying that

$$\tilde{\mathbf{G}}^l \mathbf{B} = [\mathbf{G} + vv^\top]^{l-1}[\mathbf{G} + vv^\top]\mathbf{B} = [\mathbf{G} + vv^\top]^{l-1}\mathbf{G}\mathbf{B}$$
$$= [\mathbf{G} + vv^\top]^{l-2}[\mathbf{G} + vv^\top]\mathbf{G}\mathbf{B} = [\mathbf{G} + vv^\top]^{l-2}\mathbf{G}^2\mathbf{B}$$
$$= \dots = \mathbf{G}^l \mathbf{B}.$$

Hence, we have $\mathbf{C}\mathbf{G}^l \mathbf{B} = \mathbf{C}\tilde{\mathbf{G}}^l \mathbf{B}, \ \forall l = 0, 1, \dots$ but $\mathbf{G} \neq \tilde{\mathbf{G}}$. It implies the topology $\mathbf{G}$ is not learnable according to Lemma 2.

The proof for necessity of observability of $(\mathbf{G}, \mathbf{C})$ is analogous to the above and is omitted here. ∎

**Lemma 4** *If the topology* $\mathbf{G}$ *is learnable, then* $\text{rank}(\mathbf{B}) = n$ *or* $\text{rank}(\mathbf{C}) = n$.

**Proof:** Suppose that $\text{rank}(\mathbf{B}) < n$ and $\text{rank}(\mathbf{C}) < n$. Then there exist nonzero vectors $v, u \in \mathbb{R}^n$ such that $\mathbf{C}v = 0$ and $u^\top \mathbf{B} = 0$. Without loss of generality, we assume $u^\top v \neq -1$. Next, we define a matrix $A := I + vu^\top$. Its inverse $A^{-1}$ exists according to Sherman-Morrison formula with $A^{-1} = I - \frac{vu^\top}{I + u^\top v}$.

Given a topology matrix $\mathbf{G}$, we can construct $\tilde{\mathbf{G}} := A^{-1}\mathbf{G}A$, which satisfies $\mathbf{C}\mathbf{G}^l \mathbf{B} = \mathbf{C}\tilde{\mathbf{G}}^l \mathbf{B}$ for all $l = 0, 1, \dots$. Then, according to Lemma 2 the topology $\mathbf{G}$ is not learnable. ∎

**Proof of Theorem 4:** The transfer function $F_{\mathbf{G}}(z)$ can be expanded as follows:

$$F_{\mathbf{G}}(z) = \mathbf{C}(zI - \mathbf{G})^{-1}\mathbf{B} = \mathbf{C}z^{-1}\sum_{k=0}^{+\infty}(z^{-1}\mathbf{G})^k \mathbf{B}.$$

Hence, (17) in Lemma 2 can be equivalently stated as $F_{\mathbf{G}}(z) = F_{\tilde{\mathbf{G}}}(z)$, which concludes the learnability of $\mathbf{G}$ in terms of the transfer function.

Next, we prove the learnability condition in terms of matrices $\mathbf{G}, \mathbf{B}$ and $\mathbf{C}$.

**Sufficiency:** Suppose $\text{rank}\,\mathbf{C} = n$ and $(\mathbf{G}, \mathbf{B})$ is controllable. It is sufficient to prove that for a pair of parameter values $(\mathbf{G}, \tilde{\mathbf{G}})$ $F_{\mathbf{G}}(z) = F_{\tilde{\mathbf{G}}}(z)$ holds if and only if $\mathbf{G} = \tilde{\mathbf{G}}$.

Consider two topology matrix $\mathbf{G}, \tilde{\mathbf{G}}$ with difference $\Delta := \mathbf{G} - \tilde{\mathbf{G}}$. Suppose $F_{\mathbf{G}}(z) = F_{\tilde{\mathbf{G}}}(z)$. As $\mathbf{C}$ has full column rank, we have

$$(zI - \tilde{\mathbf{G}})\mathbf{C}^{\dagger}[F_{\mathbf{G}}(z) - F_{\tilde{\mathbf{G}}}(z)] = \Delta(zI - \mathbf{G})\mathbf{B} = \sum_{k=0}^{\infty} z^{-(k+1)}\Delta\mathbf{G}^k\mathbf{B} \equiv 0.$$

As $z^{-(k+1)}$ is nonzero, it is derived from the above that

$$\Delta\mathbf{G}^k\mathbf{B} \equiv 0, \ \forall k = 0, 1, \dots \tag{20}$$

Since $(\mathbf{G}, \mathbf{B})$ is controllable, (20) holds if and only if $\Delta$ is zero matrix, which shows $\mathbf{G} = \tilde{\mathbf{G}}$.

The proof for case 2 (i.e., full row rand of matrix $\mathbf{B}$ and observability of $(\mathbf{G}, \mathbf{C})$) is analogous to the above and is omitted here.

**Necessity:** These can be obtained directly from Lemmas 3 and 4. $\blacksquare$

## F  BASELINE APPROACHES

### F.1  BASELINES IN SECTION 5.1

The method of stable least square (SLS) is defined by the optimization problem

$$\inf_{\mathbf{G} \in \mathcal{S}} \frac{1}{2}||Z - \mathbf{G}Y - \mathbf{B}V||_F^2. \tag{21}$$

The method of stable $L_2$-regularized least square (SL2LS) solves

$$\inf_{\mathbf{G} \in \mathcal{S}} \frac{1}{2}||Z - \mathbf{G}Y - \mathbf{B}V||_F^2 + \beta||\mathbf{G}||_F^2. \tag{22}$$

Solving Eq. (21) and (22) is numerically challenging. We propose to solve $\hat{\mathbf{G}}_{\mathsf{SLS}} = \arg\min_{\mathbf{G}} \frac{1}{2}||Z - \mathbf{G}Y - \mathbf{B}V||_F^2$ and $\hat{\mathbf{G}}_{\mathsf{SL2LS}} = \arg\min_{\mathbf{G}} \frac{1}{2}||Z - \mathbf{G}Y - \mathbf{B}V||_F^2 + \beta||\mathbf{G}||_F^2$ and project them to their nearest stable ones by applying Eq. (10).

### F.2  BASELINE IN SECTION 5.2

The method of stable subspace identification (SSubl) is posed by solving the following problem

$$\inf_{\mathbf{G} \in \mathcal{S}} \sum_{l=1}^{n-2} ||\mathbf{C}\mathbf{G}^l\mathbf{B} - \mathbf{M}_l^{\star}||_F^2. \tag{23}$$

We first compute $\hat{\mathbf{G}}_{\mathsf{SSubl}} = \arg\min_{\mathbf{G}} \sum_{l=1}^{n-2} ||\mathbf{C}\mathbf{G}^l\mathbf{B} - \mathbf{M}_l^{\star}||_F^2$ and then follow Eq. (10) to project it to its nearest stable matrix.

