# OpenReview forum: "Network Learning in Quadratic Games from Fictitious Plays"
_ICLR.cc/2022/Conference — ICLR 2022 Submitted_

### Official Review · Reviewer_2igb · 2021-11-02

**Correctness:** 4
**Technical Novelty And Significance:** 3
**Empirical Novelty And Significance:** Not applicable
**Recommendation:** 6
**Confidence:** 2

**Main Review:**

Strengths: The model is quite interesting and I think the model has a lot of potential. The methods used are non-trivial and the results are quite surprising and general.



Weaknesses: The paper is not very accessible.
How can all components of G be uniquely determined? That is, how can one distinguish between G_1  with g_{1,2}=5 and all other entries are say 1 or 10000 and G_2 which is the same but g_{1,2}=4.999?

Why was the name controllable chosen?
What do you mean by the indexes are sorted (besides the obvious)?
The authors spend no time giving intuition behind the theorems and since the proofs are not part of the paper, I cannot vouch for the correctness.
What does a "trajectory" mean in your setting?
Notation (G, vector)
Notation Im(B) unclear

**Summary Of The Paper:**

The authors study a game theory problem in linear quadratic games (family of payoffs) with the following assumptions
1) players are assumed to use the best-response strategy
2) an adversary can observe a subset of the nodes' actions
3) an adversary can modify the wards for a subset of the players
4) one part of the payoff function that is governed by a matrix G and models the non-marginal payoffs i.e., the part of the payoff that is a function of the actions taken by the other agents.


The authors study under what circumstances G can be learned.
One result is that that under some mild assumptions when the adversary can observe all players agents, then G can be learned.
The authors show for more settings necessary and sufficient conditions for learning G. The authors also perform some simulations on random graphs.

**Summary Of The Review:**

Seems like a very nice paper, but not accessible enough for me to understand all the details.

---

### Official Review · Reviewer_Zoba · 2021-11-02

**Correctness:** 2
**Technical Novelty And Significance:** 2
**Empirical Novelty And Significance:** 2
**Recommendation:** 3
**Confidence:** 4

**Main Review:**

- Strengths:
  - Learning the structure of games of this type has attracted some attention recently and using fundamental concepts from control theory (observability and controllability of a linear system) is an interesting view of this problem.
 -  The fact that the adversary may have a partial observation of the actions of the agents can be quite relevant in practice and does not seem to have been explored in previous work.

- Weaknesses: The paper has some technical issues; more specifically :
   - The title of the paper is misleading, as the paper examines the case where all players play the best response (Fictitious play usually denotes a strategy where each player plays the best response to the "average" strategy observed from her opponents).
  -  The setting is very specific, as only best response dynamics are treated (in principle, agents may use other learning algorithms). In addition, the authors mention that "In practice, it is expected that the best response dynamics (BR) should converge to a Nash equilibrium", however this is generally not the case (even if the NE of a game is unique) - the authors should provide a reference or prove that in this particular game BR converges to a NE.
   -  The ability of the adversary to affect / perturb the agents' actions seems arbitrary and needs to be motivated. In addition, the effect of this ability is never really explored in the paper (can it, for example, offset partial observability?).
   -  Similarly, the fact that $G$ is usually sparse is nowhere motivated.
   -  Theorem 2 seems to hold only for the case where the adversary can also purturb the actions of all agents (in addition to having full observability).
   -  Theorem 4 seems just a direct application of van der Waarde et al. (2021)

- Other comments:
   - The authors mention that they consider a "setting of dynamic game" ("Related work" paragraph, Section 1.4), however this is not true; the game under consideration is a repeated game, where agents follow best response dynamics.
   -  It is unclear what is the connection of Sections 3 and 4 to Section 2: the theorems proved in the latter section do not seem to be used in any way.
   -  In Section 4.2 it is mentioned that the observation of the actions of the players is "masked by noise" - however, noisy observations are not mentioned before.

**Summary Of The Paper:**

The paper addresses the problem of an adversary learning the structure (i.e. coupling matrix $G$ among agents in the payoff) of a repeated game with quadratic payoffs where the agents are playing under best response dynamics and the adversary can potentially observe the actions of only a subset $O$ of players and also affect the action of another subset $M$ of players. The authors give conditions for which $G$ can be recovered, depending on the sets $O$ and $M$, when a single trajectory or multiple trajectories of the game are available. In addition, they give algorithms for the cases where all and part of the actions are observable and $G$ is a sparse matrix.

**Summary Of The Review:**

Application of concepts of control theory in learning the structure of a game with linear-quadratic payoff may be interesting, especially in the case of partially observable actions, however the setting of this paper seems specific without being very well motivated, there are some technical errors and not well supported arguments and the paper reads somewhat incoherent.

---

### Official Review · Reviewer_YLLb · 2021-11-07

**Correctness:** 2
**Technical Novelty And Significance:** 2
**Empirical Novelty And Significance:** 2
**Recommendation:** 3
**Confidence:** 4

**Main Review:**

I do not believe the paper is a good fit for ICLR for the following reasons:

1. The authors' model is highly stylized and I was unable to see any connection to the type of applications and/or theoretical questions that are relevant to ICLR. [For instance, the role of the "overseeing adversary" is never explained, nor the player dynamics, nor the specific game model] Even though the paper cites some papers that have appeared in ML venues and seem to be relevant, there is no denying that the core of the paper lies in control theory (or, possibly, econometrics). In particular, the authors do not attempt to provide _any_ intuition or motivation for many of the notions that they employ (for instance, the notion of "controllability"), so the paper would be inaccessible to the wider ICLR community.

1. In addition to the above, the paper suffers from imprecise writing, to the extent that the main theorems are impossible to parse. For example, Theorems 1 and 2 state that "$G$ can be uniquely determined by the adversary for sufficiently large $T$" if and only if a certain condition holds. But what does "uniquely determined" mean in this context? What algorithm is the adversary following? One of the algorithms described in Sections 3 and 4? If so, the corresponding theorems should be stated as guarantees for the algorithms in question (and the paper completely rewritten accordingly); if not, what is the notion of "learnability" that the authors refer to, and what is the role of $T$? Likewise, what does "access to infinitely many trajectories" mean? The adversary is assumed to be able to maintain a variable with infinite memory?

1. The authors make no attempt to explain whether the conditions provided in Theorems 1-4 are light or stringent, and they likewise provide no intuition as to what they mean for the underlying game. The notion of controllability is classical in control theory, but this does not shed any insight on what this actually means for the system at hand, and the authors make no effort to explain any of this. [Incidentally, this further reinforces my belief that this paper is more appropriate for a control venue like CDC, and not ICLR.]

I do not think that the above can be fixed with a revision in a few days, hence my "strong reject" score. The authors should not interpret this as a critique on the worth of their results, but as an indication of (a) the suitability of this paper to the ICLR community at large; and (b) the level of rewriting that would be required to make the paper accessible to said community in the first place.
### Specific remarks

I am providing below a list of detailed remarks that could help the authors in an eventual resubmission /revision of the paper (irrespective of venue):

1. The use of the term "fictitious play" is erroneous. Fictitious play means best-responding to the empirical frequency of the opponents' play, not to their last action (see the original papers by Brown and Robinson in the 50's). The (unperturbed) process considered by the authors is the best response dynamics considered by, e.g., Monderer and Shapley (1996), not fictitious play.

1. The plural of "equilibrium" is "equilibria", not "equilibriums".

1. The authors never define $g_ii$. Is it allowed to take any value? Is it assumed that the sum in (1) is only taken over $j\neq i$ (which would give $g_{ii} = -1/2$)?

1. The authors state in the beginning of Section 3 that the best response dynamics are expected to converge to Nash equilibrium, and that this is the case if $\rho(G) < 1$ (Ballester et al., 2006). The authors subsequently seem to suggest that this occurs "in practice" because $G$ is sparse but I do not see what this has to do with the spectral radius of $G$: the matrix $g_{1,n} = g_{n,1} = 1$ and $g_{ij}=0$ otherwise is as sparse as it can get, but its spectral radius is $1$.

1. What does "random generate $\delta$" mean in Algorithm 1? And if the algorithm is stochastic, how does it connect to the (presumably) deterministic Theorems 1-4?

1. Line 1 in Algorithm 2 is also problematic: what does "carry out classical subspace identification" mean, and how is this implemented?



**Summary Of The Paper:**

In this paper, the authors consider quadratic network games with payoff functions of the form
$$
J_i(x_i;x_{-i}) = \alpha_i x_i - x_i^2/2 + \sum\nolimits_{j=1}^{n} x_i x_j
$$
where $\alpha_i>0$ denotes the marginal benefit of the $i$-th player from playing $x_i \in \mathbb{R}$, and $g_{ij}$ is a matrix of interactions that determines whether players $i$ and $j$ are adversaries ($g_{ij}<0$), friends ($g_{ij}>0$), or non-interacting ($g_{ij}=0$).

The authors assume that players follow a "perturbed" best-response model of the form
$$
x_i(t+1) = \arg\max\nolimits_{x_i\in\mathbb{R}} J_i(x_i;x_{-i}(t)) + u_i(t+1)
$$
where $u_i$ is a perturbation that affects only a subset $M$ of "manipulable" players (possibly empty).

The authors are interested in conditions under which the underlying interaction matrix $G=(g_{ij})_{i,j=1,\dots,n}$ can be learned from the outcome of the players' learning process. In this regard, they provid the following results:

1. If all player actions are observable, $G$ is learnable from a single trajectory of play if and only a certain matrix involving $x(1) - x(0)$ is controllable (Theorems 1 and 2; the condition in Theorem 2 involves a non-empty set of manipulable players and is less stringent).

1. If only a subset of players are observable, $G$ is learnable from a single trajectory of play if and only if a certain Markov equation admits a unique solution (Theorem 3).

1. Finally, if the learner has access to infinitely many trajectories, $G$ is learnable if and only if two pairs of matrices are controllable / observable.

Two algorithms for identifying $G$ are described in Sections 3 and 4 with full partial action observations respectively (though it was not clear to me what the exact relation is with respect to Theorems 1-4). Finally, in Section 5, the authors provide a series of numerical validation results over different random network structures (Erdős-Rényi, Barabasi-Albert,etc.).




**Summary Of The Review:**

I do not believe the paper is a good fit for ICLR for the following reasons:

1. The authors' model is highly stylized and there is no connection to the type of applications and/or theoretical questions that are relevant to ICLR.

1. The paper suffers from imprecise writing, to the extent that the main theorems are impossible to parse.

1. The authors make no attempt to explain their results and the connections of the required conditions to the underlying game.

This paper could be a good fit to a control theory conference like CDC, but not ICLR.

---------------
Post-rebuttal
---------------
I have read the answers of the authors.  Unfortunately, I am not convinced and I will retain my original score.

---

### Author Response · Authors · 2021-11-22
**Reply to Reviewer YLLb's Comments**

We thank the reviewer for the time in evaluating our work as well as the useful comments. \
      In response to the concerns raised in Reviewer YLLb's comments, we first clarify the connection of our work with ICRL: \
      1) The specific game model we consider is the linear quadratic game, which is a classic type of network games, with an underlying graph structure characterizing the players' inter-personal influences. Despite its simplicity, the linear quadratic game has been a fundamental model to describe many practical situations, e.g., the user behaviors for online e-commerce platforms in Leng et al. (2020). \
      2) In computer science literature, network games are known as graphical games (Kearns et al., 2001), and learning the graph structure (or relationship among entities) from observed action data is a direction investigated in the machine learning communities. There have been a few studies recently. For example, the works in Irfan & Ortiz (2014); Honorio & Ortiz (2015); Ghoshal & Honorio (2016; 2017) have proposed to learn graphical games by observing actions from linear influence games with linear influence functions. Especially, Leng et al. (2020) in ICML proposed algorithms to learn graph structure of the linear quadratic game given the Nash equilibrium (NE) action of the games. Our work relaxes this condition and discuss a different case when NE is not available but the observations from players' dynamics is provided. Moreover, to explore the information of graph structure from the observations, we borrow the tools from control theory, and hence some classic concepts in control theory (such as, controllability) are introduced in the manuscript. \
	In summary, the game model discussed in our work has practical applications, and the problem of learning graph structure is of interest to audience from ICRL communities, even though some control theoretic concepts are utilized to investigate the learning problem. \
        Second, we explain some concepts mentioned in the reviewer YLLb's comments: \
	1. "uniquely determined" indicates that solution uniqueness of computing the graph structure $G$ from the information set $\mathcal{I}$ (defined in page 2, Section 1.3). Moreover, "learnability" is defined in Section 1.3, page 2, and term $T$ reflects the size of information set (or size of data set in machine learning community), which will affect the learnability and further the learning accuracy of $G$.
	2. Section 2 investigates the theoretical identifiability of graph structure $G$, in other words, is it possible to compute $G$ when the attacker has performance knowledge of the information set $\mathcal{I}$. It indicates the fundamental limitation of learning performance from the perspective of attacker. Results in Theorems 1-4 are unrelated with the specific algorithm used by the attacker.
	3. "access to infinitely many trajectories" does not imply that the attacker must have infinite memory. It is a condition to support our theoretic results. The specific algorithms are provided afterwards (Sections 3-4) which instruct the attacker how to learn the graph structure $G$ practically. To implement these algorithms, finite memory is enough for the attacker. \
	4. we acknowledge that the current version lacks some explanations about the underlying intuition. More explanation will be added in the next step. \
        Last, we explain the specific remarks mentioned by the reviewer: \
        5.1, we agree with the reviewer about the terms "fictitious play" and "equilibria", and the manuscript will be revised carefully in accordance to these comments. \
        5.2, according to the standard definition of linear quadratic game, the underlying graph structure does not consider self-loop, in other words, $g_{i,i}=0$. \
        5.3, the sparsity assumption is unrelated with the spectrum assumption of $\rho(\mathbf{G})<1$. They are two separate assumptions on the graph structure $\mathbf{G}$. \
        5.4, the term $\delta$ is generated randomly, e.g., a random variable following Gaussian distribution. The proposed algorithm and the theoretic results focus on different parts of the learning problem. But they are complementary and consist a comprehensive study of learning the graph structure from data observations. \
        5.5, we apologize for the missing reference of ``classic subspace identification method" in Algorithm 2. There are lots of different subspace identification methods in the literature, and please see more implementation details in the book Van Overschee & De Moor (2012).

---

### Author Response · Authors · 2021-11-22
**Reply to Reviewer Zoba's Comments**

We thank the reviewer for the time in evaluating our work as well as the useful comments. We agree with the reviewer Zoba about the terms "fictitious play", "dynamic game", and the manuscript will be revised carefully in accordance to this comment. Reply to other comments: \
       1. Regarding the specific setting, first, the response dynamics is a good start to investigate the problem of learning underlying graph structure given observations from the players' dynamics. Even though the form of best response is simple, the learning problem is the opposite, on which our work provides a comprehensive analysis. Second, the best response will converge to NE, which can be proved as follows. The dynamics of $x_i(k)$ (BR) can be rewritten in a compact form: $x(k+1) = \alpha + G x(k)$ where $x(k) := [x_1(k),...,x_n(k)]$. Set $x(0)=0$, and then $x(k+1)=\sum_{t=0}^{k} G^{t}\alpha$, which will converge to the NE (i.e., $(I-G)^{-1}\alpha$) if the spectrum radius of the matrix $G$ is less than 1. \
        2. Regarding the ability of attacker, it can: a). select a subset of agents to inject perturbation; b). design the injected perturbation (note that value of the perturbation could be arbitrary). In this sense, Theorems 1-4 can be regarded as a description of the adversary's ability, that is, the graph structure can be uniquely determined/computed if the perturbed agents and the injected perturbation satisfy certain conditions. Moreover, the unobservability is the consequence caused by which subset is chosen by the adversary. \
        3. The sparsity of graph $G$ is motivated by many practical situations. Sparsity is a fundamental characteristic of numerous biological, social, and technological networks. See references Cassidy & Solo (2014), Wisitpongphan et al. (2007), Bao and Mokbel (2012). \
        4.The condition (i) in Theorem 2 is unrelated with the location of perturbed agents, and hence we discuss whether condition (ii) holds for a subset pf perturbed agents. Consider a graph structure with 4 nodes, and the matrix G is $[0, 0, 0, 0; g_{2,1}, 0, 0, 0; g_{3,1}, 0, 0, g_{3,4}; g_{4,1}, 0, 0, 0]$. The matrix B is $[b_1, 0; 0, b_2; 0, 0; 0, 0]$, which means that only players 1 and 2 are perturbed. It can be proved that the pair $(G,B)$ is controllable according to the definition in Section 2.1, page 4. See more details of this example on website: https://en.wikipedia.org/wiki/Network_controllability. \
       5. We acknowledge that results in Theorem 4 relies on Van der Waarde et al. (2021), which has already been indicated in the footnote 1, page 14. The main difference between our approach and Van der Waarde et al. (2021) lies in the proof details. Here, we established the equivalence between Markov parameters and indistinguishability of $G$ in lemma 2, which has been used to greatly simplify the proof of Theorem 4. \
       6. This work investigates the problem of learning the graph structure $G$ from observations of players' dynamics. On one hand, Section 2 investigates the theoretical identifiability of graph structure $G$, in other words, is it possible to compute $G$ when the attacker has performance knowledge of the information set $\mathcal{I}$. On the other hand, Sections 3-4 provide specific algorithms which instruct the attacker how to learn the graph structure $G$ practically. In summary, these two parts are complementary and consist a comprehensive study of the learning problem.

---

### Author Response · Authors · 2021-11-22
**Reply to Reviewer 2igb's Comments**

We thank the reviewer for the time in evaluating our work as well as the useful comments. Regarding to the comments from Reviewer 2igb, \
       1. "uniquely determined" indicates that solution uniqueness of computing the graph structure $\mathbf{G}$ from the information set $\mathcal{I}$ (defined in page 2, Section 1.3). It characterizes the fundamental limitation of attacker's learning performance from the theoretic perspective. Obviously, if the uniqueness fails, the two graph structures mentioned in reviewer's comments can not be distinguished. \
	2. the name "controllable" is a classic concept in control theory. \
	3. we explain "the indexes are sorted" by the following example: If this game has three action-compromised players (i.e., players 1, 3, 6), then $m=3$ and $M=\{p_1 =1, p_2=3, p_3 = 6\}$. \
       4. a "trajectory" means one realization of the information set $\mathcal{I}$ (defined in page 2, Section 1.3). \
       5. we acknowledge that the current version lacks some explanations about the underlying intuition. More explanation will be added in the next step.

---

> ### Comment · Reviewer_2igb · 2021-11-25
> **ACK**
>
> Thank you for the response. My evaluation remains the same.

---

### Author Response · Authors · 2021-11-22
**References for Replies**

References: \
Kearns, M., Littman, M., and Singh, S. Graphical models for game theory. In Proceedings of the 17th Conference on Uncertainty in Artificial Intelligence, 2001. \
\
Irfan, M. and Ortiz, L. On influence, stable behavior, and the most influential individuals in networks: A game-theoretic approach. Artificial Intelligence, 215:79–119, 2014. \
\
Honorio, J. and Ortiz, L. E. Learning the structure and parameters of large-population graphical games from behavioral data. Journal of Machine Learning Research, 16: 1157–1210, 2015. \
\
Ghoshal, A. and Honorio, J. From behavior to sparse graphical games: Efficient recovery of equilibria. In Proceedings of the IEEE Allerton Conference on Communication, Control, and Computing, 2016. \
\
Ghoshal, A. and Honorio, J. Learning graphical games from behavioral data: Sufficient and necessary conditions. In Proceedings of the 20th International Conference on Artificial Intelligence and Statistics, pp. 1532–1540, 2017. \
\
Leng, Y., Dong, X., Wu, J., and Pentland, A. Learning Quadratic Games on Networks. In International Conference on Machine Learning, pp. 5820-5830, 2020. \
\
Peter Van Overschee and BL De Moor. Subspace identification for linear systems: Theory—Implementation—Applications. Springer Science & Business Media, 2012.  \
\
Cassidy, B., Rae, C. and Solo, V., Brain activity: Connectivity, sparsity, and mutual information. IEEE transactions on medical imaging, 34(4), pp.846-860, 2014. \
\
Wisitpongphan, N., Bai, F., Mudalige, P., Sadekar, V. and Tonguz, O., Routing in sparse vehicular ad hoc wireless networks. IEEE journal on Selected Areas in Communications, 25(8), pp.1538-1556, 2007. \
\
Bao, J., Zheng, Y. and Mokbel, M.F., Location-based and preference-aware recommendation using sparse geo-social networking data. In Proceedings of the 20th international conference on advances in geographic information systems (pp. 199-208), November, 2012.

---

### Decision · Program_Chairs · 2022-01-20

**Decision:**

Reject

**Comment:**

In this paper, the authors consider linear quadratic network games (also known as graphical games) and they discuss a number of conditions and procedures to learn the underlying graph of the game from observations of best-response trajectories (or possibly infinite sets thereof) in the game.

The reviewers' initial assessment was overall negative, with two reviewers recommending rejection and one giving a borderline positive recommendation. The authors' rebuttal did not address the concerns of the reviewers recommending rejection, and the authors did not provide a revised paper for the reviewers to see how the authors would implement the suggested changes, so the overall negative assessment remained.

After my own reading of the paper, I concur with the majority view that the paper has several weaknesses that do not make it a good fit for ICLR (especially regarding the lack of precision in the theorems and the statement of the relevant assumptions), so I am recommending rejection.